# Inhibition of Prostate Cancer Cell Survival and Proliferation by Carnosic Acid Is Associated with Inhibition of Akt and Activation of AMPK Signaling

**DOI:** 10.3390/nu16091257

**Published:** 2024-04-24

**Authors:** Matteo Nadile, Newman Siu Kwan Sze, Val A. Fajardo, Evangelia Tsiani

**Affiliations:** 1Department of Health Sciences, Brock University, St. Catharines, ON L2S 3A1, Canada; mn17fp@brocku.ca (M.N.); nsze@brocku.ca (N.S.K.S.); vfajardo@brocku.ca (V.A.F.); 2Centre for Bone and Muscle Health, Brock University, St. Catharines, ON L2S 3A1, Canada

**Keywords:** prostate cancer, carnosic acid, survival, proliferation, Akt, AMPK, Sestrin-2

## Abstract

Prostate cancer, accounting for 375,304 deaths in 2020, is the second most prevalent cancer in men worldwide. While many treatments exist for prostate cancer, novel therapeutic agents with higher efficacy are needed to target aggressive and hormone-resistant forms of prostate cancer, while sparing healthy cells. Plant-derived chemotherapy drugs such as docetaxel and paclitaxel have been established to treat cancers including prostate cancer. Carnosic acid (CA), a phenolic diterpene found in the herb rosemary (Rosmarinus officinalis) has been shown to have anticancer properties but its effects in prostate cancer and its mechanisms of action have not been examined. CA dose-dependently inhibited PC-3 and LNCaP prostate cancer cell survival and proliferation (IC_50_: 64, 21 µM, respectively). Furthermore, CA decreased phosphorylation/activation of Akt, mTOR, and p70 S6K. A notable increase in phosphorylation/activation of AMP-activated kinase (AMPK), acetyl-CoA carboxylase (ACC) and its upstream regulator sestrin-2 was seen with CA treatment. Our data indicate that CA inhibits AKT-mTORC1-p70S6K and activates Sestrin-2-AMPK signaling leading to a decrease in survival and proliferation. The use of inhibitors and small RNA interference (siRNA) approaches should be employed, in future studies, to elucidate the mechanisms involved in carnosic acid’s inhibitory effects of prostate cancer.

## 1. Introduction

Prostate cancer affects millions of people and is the second leading cause of death in men accounting for 375,304 deaths in 2020 worldwide [1,2,3]. The risk of developing prostate cancer increases with age, with more than 85% of those diagnosed being older than 60 years of age [1,3,4]. Prostate cancer is strongly associated with alterations in oncogenes and/or tumor suppressor genes [5,6]. These aberrations result in changes in gene transcription and translation of proteins involved in proliferation and cell death [7,8].

Accumulation of DNA mutations as well as epigenetic alternations in proteins, lead to uncontrolled cell proliferation and evasion of apoptosis which give rise to tumor formation and cancer [9]. Cellular signaling cascades, such as the Ras-mitogen activated protein kinase (MAPK) and the phosphatidylinositol 3-kinase (PI3K)/protein kinase B (Akt)/mechanistic target of rapamycin (mTOR)/mTORC1 complex are overactivated, contributing to increased protein synthesis, proliferation, and cell survival [10,11,12,13]. These signaling cascades are activated by growth factor receptor such as epidermal growth factor receptor (EGFR) signaling [14].

Androgens play a large role in the development as well as progression of prostate cancer. Prostate epithelial cells express high levels of androgen receptors (AR) [15], with a higher expression being in the luminal epithelial compared to the basal epithelial cells [16]. AR contains three main functional binding domains: the N-terminal transcriptional regulation domain, the DNA binding domain (DBD) and the ligand binding domain [17,18]. Androgens such as testosterone are the main ligands that bind to the ligand binding domain of ARs and form dimers in the nucleus, which bind to androgen response elements (AREs) in AR-regulated genes and upregulate their transcription leading to cell growth and proliferation, cell cycle progression, and protein synthesis [18,19,20,21,22].

Activated Akt leads to the downstream activation of the mammalian target of rapamycin (mTOR) and p70 S6 kinase (p70 S6K) [23,24], resulting in increased protein synthesis and proliferation. Deletions in phosphatase and tensin homolog (PTEN), the negative regulator of PI3K, and overactivation of Akt drives prostate cancer [25], while targeting this cascade may provide treatment benefits [26,27,28]. Other genetic aberrations seen in prostate cancer include loss of RB1 and p53, N-Myc overexpression, *AR*, *c-MYC* and FOXA1 overexpression [22,29]. These alterations lead to loss of basal cell layer, disruption of normal tissue architecture, detectable levels of prostate specific antigen (PSA) in the bloodstream, and prostate cell proliferation [15].

Adenosine monophosphate-activated protein kinase (AMPK) is a 62 kDa heterotrimeric protein with anticancer effects [30,31,32,33]. Its activation leads to the inhibition of the mTORC1 complex [34]. This is achieved through the phosphorylation of raptor, mTOR’s binding partner, at Ser722 and Ser792 [35]. AMPK activation causes the phosphorylation/inhibition of acetyl-CoA carboxylase (ACC) and 3-hydroxy-3-methyl-glutaryl-CoA reductase (HMG-CoA reductase) leading to impairment of fatty acid and sterol synthesis, respectively [36,37]. Liver kinase B1 (LKB1), Ca^2+^/calmodulin (CaM)-dependent protein kinase kinases (CaMKK) and transforming growth factor beta-(TGF-β)-activated kinase 1 (TAK1) are responsible for AMPK activation by phosphorylating the Thr172 residue [38,39,40,41]. Sestrin-2 has been implicated in AMPK activation by directly interacting with the α subunit [42,43], leading to mTOR inhibition [44].

Plant-derived chemicals serve as valuable sources for cancer treatment [45]. Paclitaxel and docetaxel, potent drugs used in the treatment of various cancers including prostate cancer, were originally isolated from the bark of the Pacific yew tree (*Taxus brevifolia*) and the needles of the European yew tree (*Taxus baccata*), respectively [46,47]. These taxanes act by hindering mitosis through microtubule binding, leading to inhibition of depolymerization and formation of mitotic spindles [48,49]. However, due to an increase in cancer cases and the growing resistance to current chemotherapy drugs, alternative treatment strategies are required.

In the past few years our lab has focused on finding novel plant-derived chemicals with anticancer potential. We observed promising effects against prostate cancer using the polyphenol resveratrol that were associated with inhibition of the Akt-mTOR and activation of the AMPK pathways [50]. More recently we examined the effects of rosemary extract (RE) [51,52,53,54,55], and RE polyphenols [56,57]. RE elicited inhibitory effects in prostate [54], breast [53] and lung cancer cells [55]. Recently, we discovered that the diterpene carnosic acid (CA), contained in RE, has significant inhibitory effect on lung cancer cell survival and proliferation [57]. Nevertheless, limited studies have examined the effects of CA in prostate cancer. Treatment of PC-3 and DU-145 androgen-insensitive prostate cancer cells with CA (6.25–100 µg/mL, 18.8–301 µM) resulted in a decrease in cell viability [58] and proliferation and induced both extrinsic and intrinsic apoptotic pathways in PC-3 cells [59]. Treatment of LNCaP, 22Rv1, PC-3 and DU145 cells with CA decreased cell viability, proliferation and induced apoptosis [60] and caused cell cycle arrest [61].

Although these limited studies have previously demonstrated the anticancer potential of CA, its effects on signaling cascades and, thus, its mechanism of action are currently unknown. In the present study, we investigated the effects of CA on prostate cancer cells. We hypothesized that CA would inhibit prostate cancer cell survival and proliferation while increasing AMPK signaling.

## 2. Materials and Methods

### 2.1. Materials

Human PC-3, LNCaP prostate cancer and PNT1A prostate epithelial cells were obtained from America Type Culture Collection (ATCC). Cell culture (RPMI) media, fetal bovine serum (FBS), trypsin, and antibiotic were from GIBCO (Burlington, ON, Canada). Antibodies that recognized total or phosphorylated (AMPK: cat. No. 5831, pAMPK (Thr172): cat. No. 2535, ACC cat. No. 3676S, pACC (Ser79) cat. No. 11881S, Akt cat. No. 9272S, pAkt (Ser473) cat. No. 9271S, mTOR Cat. No. 2972S, pmTOR Ser2448 Cat. No. 2971S, Raptor Cat. No. 2280S, pRaptor Ser792 Cat. No. 2083S, and p70 S6K Cat. No. 9202S, pp70 S6K Thr389 Cat. No. 9205S, Sestrin-2 cat. No. 8487, vinculin Cat. No. 4650S and β-actin cat. No. 4967 were purchased from Cell Signaling Technology via New England Biolabs (Mississauga, ON, Canada). Bovine serum albumin, dimethyl sulfoxide (DMSO), methylene blue and crystal violet stain, and docetaxel were purchased from Millipore Sigma (Oakville, ON, Canada). Carnosic acid was purchased from MedChemExpress in Monmouth Junction, NJ, USA.

### 2.2. Cell Culture and Treatment

PC-3, LNCaP and PNT1A cells were grown in RPMI media supplemented with 10% (*v*/*v*) FBS, and 1% (*v*/*v*) antibiotic-antimycotic solution in a humidified atmosphere of 37 °C at 5% CO_2_. Initial stock solutions of CA (100 mM) and DTX (10 mM) were prepared in DMSO followed by working solutions using cell culture media. The time of exposure and concentration of carnosic acid (CA), and docetaxel (DTX) are indicated in each figure.

### 2.3. Clonogenic Survival Assay

Clonogenic survival assays were performed as previously described [51]. Cells were seeded at a density of 1000 cells/well in a 6-well plate and were allowed to adhere overnight. Cells were incubated with indicated concentrations of CA for 7 days. Following 7 days of treatment, cells were washed twice and stained with 0.05% (*w*/*v*) methylene blue dye. The next day, the cells were counted and colonies with >50 cells were recorded.

### 2.4. Cell Proliferation Assay

The crystal violet cell proliferation assay was performed as described previously [62]. Cells were seeded (4000 cells/well) in sextuplicate in 96-well plates and treated with the indicated concentration of CA for 24 or 72 h. Cells were fixed with 10% formalin and stained using 0.5% crystal violet stain following a 24 or 72 h treatment. The plate was allowed to dry, and cells were solubilized, and the absorbance was measured at 570 nm KC4 plate reader (Bio-Tek, Winooski, VT, USA). The data are expressed as percent of control.

### 2.5. Immunoblotting

Immunoblotting was performed as previously described [62]. Cells were seeded and allowed to grow to 90% confluence. Following their treatment cells were washed with ice-cold PBS and then lysed with ice-cold lysis buffer. Lysates were collected and 5% β-mercaptoethanol containing SDS buffer was added and boiled for 5 min. 20 µg protein samples were separated using SDS-PAGE, transferred on PVDF membrane, and incubated with primary antibody buffer overnight at 4 °C and the following day the membranes were incubated with horse radish peroxidase (HRP)-linked IgG anti-rabbit secondary antibody for 1 h at room temperature before being visualized (using BioRad chemidoc imager). Signals were detected using Bio-Rad Clarity Western ECL Solution. Densitometric analysis was performed using Image J 1.54g software and expressed relative to the control group.

### 2.6. Statistical Analysis

All results are expressed as the mean of several individual experiments ± standard error of the mean (SEM). Significance testing was done using Graphpad Prism 9 software to perform analysis of variance (ANOVA). Significant ANOVA results were followed by Dunnett’s post hoc test. *p*-values less than 0.05 were considered significant.

## 3. Results

### 3.1. Carnosic Acid Inhibits Prostate Cancer Cell Survival

PC-3 prostate cancer cells were treated with 5, 10, 15, 30, or 60 µM of CA and the ability of cells to survive and form colonies was examined by performing a clonogenic survival assay [53]. Carnosic acid dose-dependently decreased cell survival. A significant (*p* < 0.0001) decrease in cell survival was seen at a concentration as low as 5 µM and complete inhibition of cell survival was seen at and above 30 µM (CA 5 µM: 51.6 ± 5.5%, *p* < 0.0001; CA 10 µM: 21.3 ± 4.9%, *p* < 0.0001; CA 15 µM: 11.6 ± 4.5% of control, *p* < 0.0001; Figure 1B). In addition, we used LNCaP prostate cancer cells and treated them with 2.5, 5, 10, 15, or 20 µM of CA. We found a dose-dependent decrease in cell survival with significant inhibition seen at a concentration as low as 2.5 µM and complete inhibition seen at 15 µM. (CA 2.5 µM: 61.9 ± 1.9%, *p* < 0.0001; CA 5 µM: 24.2 ± 1.9%, *p* < 0.0001; CA 10 µM: 2.8 ± 1.0% of control, *p* < 0.0001; Figure 1D).

### 3.2. Carnosic Acid Inhibits Prostate Cancer Cell Proliferation

Cells were treated with 5, 10, 20, 40, 60, 80, 100, 125, or 150 µM of CA and proliferation was assessed using the crystal violet assay. Following treatment with CA; PC-3, and LNCaP cell proliferation decreased in a dose-dependent manner with a calculated IC_50_ value of 64 and 21 µM, respectively. For PC-3 cells, CA concentrations ranging from 100–150 µM resulted in maximum inhibition of proliferation (CA 40 µM: 76.8 ± 6.7%, *p* < 0.001; CA 60 µM: 62.0 ± 5.1%, *p* < 0.0001; CA 80 µM: 32.2 ± 3.1%, *p* < 0.0001; CA 100 µM: 17.35 ± 1.9%, *p* < 0.0001; CA 125 µM: 13.9 ± 0.8%, *p* < 0.0001; CA 150 µM: 14.6 ± 1.0% of control, *p* < 0.0001; Figure 2A (blue trendline)). For LNCaP cells, CA concentrations ranging from 80–150 µM resulted in maximum inhibition of proliferation (CA 10 µM: 76.7 ± 4.4%, *p* < 0.01; CA 20 µM: 53.8 ± 6.1%, *p* < 0.001; CA 40 µM: 25.7 ± 2.3%, *p* < 0.001; CA 60 µM: 15.3 ± 2.1%, *p* < 0.0001; CA 80 µM: 9.8 ± 3.4%, *p* < 0.0001; CA 100 µM: 8.9 ± 3.6%, *p* < 0.0001; CA 125 µM: 9.4 ± 3.6%, *p* < 0.0001; CA 150 µM: 9.2 ± 3.7% of control, *p* < 0.0001; Figure 2A (red trendline)).

PNT1A cells, which represent normal prostate epithelium were treated with the same concentrations of CA for 24 h and found an inhibitory effect with an IC_50_ of 139.4 µM (Figure 2B), a much higher concentration compared to the IC_50_ for PC-3 and LNCaP prostate cancer cells.

### 3.3. Carnosic Acid Inhibits Akt Signaling

Once it was identified that survival and proliferation of PC-3 prostate cancer cells was inhibited by CA, Akt signalling was investigated since it plays a major role in protein synthesis, cell proliferation and survival [63]. Treatment of PC-3 cells with CA resulted in decreased phosphorylation of Akt Ser473 residue, an established marker of its activation [64]. A significant inhibition was seen after both 24 and 48 h of CA treatment (CA 24 h: 32.9 ± 16.9%, *p* < 0.05; CA 48 h: 5.5 ± 3.3% of control, *p* <0.001; Figure 3). Interestingly, treatment with docetaxel (DTX) (10 nM, 48 h) an established chemotherapeutic drug had no significant effect of Akt phosphorylation/activation.

### 3.4. Carnosic Acid Inhibits mTORC1-p70 S6K Signaling

mTOR and p70 S6K the downstream targets of Akt were also examined as they are key players in protein synthesis and cell proliferation signalling. Cells treated with CA showed a significant decrease in mTOR phosphorylation/activation (CA 24 h: 45.5 ± 10.3%, *p* < 0.01; CA 48 h: 32.0 ± 7.8% of control, *p* < 0.001; Figure 4A). The mTORC1 complex consists of mTOR and the regulatory protein, raptor, allowing it to interact with downstream targets [65]. Cells treated with CA showed a significant increase in raptor phosphorylation (CA 24 h: 617.7 ± 146.5%, *p* < 0.01; CA 48 h: 544.0 ± 106.2% of control, *p* < 0.01; Figure 4B). A decrease in p70 S6K phosphorylation/activation was also seen (CA 24 h: 24.9 ± 23.3%, *p* < 0.01; CA 48 h: 2.1 ± 1.2% of control, *p* < 0.001; Figure 4C).

### 3.5. Carnosic Acid Activates AMPK Signaling

Previous studies by our group found a robust phosphorylation/activation of AMPK by CA in muscle [66], fat [67] and lung cancer cells [57] and since AMPK plays a key role in energy homeostasis [68] and its activation may lead to anticancer effects [69,70], we examined the effects of CA on prostate cancer cell AMPK and its downstream target ACC. Treatment of PC-3 cells with CA significantly increased AMPK phosphorylation/activation (CA 24 h: 816.2 ± 167.0%, *p* < 0.01; CA 48 h: 1997.9 ± 314.9% of control, *p* < 0.0001; Figure 5A) and phosphorylation of ACC (CA 24 h: 806.7 ± 217.8%, *p* < 0.01; CA 48 h: 936.3 ± 269.3% of control, *p* < 0.001; Figure 5B).

### 3.6. Carnosic Acid Activates Upstream Regulators of AMPK Signaling

In an attempt to understand the mechanism of AMPK activation by CA, we examined sestrin-2 which is known to have a direct effect of activating AMPK [71,72]. We found a significant increase in sestrin-2 levels with CA (CA 24 h: 162.7 ± 9.2%, *p* < 0.05; CA 48 h: 236.3 ± 38.27% of control, *p* < 0.001; Figure 6), whereas treatment with DTX had no effect on in Sestrin-2 levels.

### 3.7. AMPK Activation; Akt and mTOR Inhibition Mimic the Effects of Carnosic Acid

We used the established AMPK activator, 5-Aminoimidazole-4-carboxamide ribonucleotide (AICAR) (500 µM), the Akt inhibitor, wortmannin (1 µM), and the mTOR inhibitor, rapamycin (200 nM) and examined and compared their effects to the effects of CA treatment. We found significant inhibition of PC-3 prostate cancer cell proliferation with these treatments that were comparable to those achieved with CA treatment (Figure 7).

## 4. Discussion

In previous studies from our lab we found that rosemary extract (RE) inhibited survival and proliferation of prostate [54], breast [53] and lung cancer cells [55]. In recent studies we focused on the diterpene CA, contained in RE and found significant inhibition of lung cancer cell survival and proliferation [57]. In the present study, we examined the effects of CA on prostate cancer cells. Our data show that treatment of PC-3 prostate cancer cells with CA resulted in inhibition of survival and proliferation. These data are in agreement with the findings of other studies by Yesil-Celiktas et al. [58], Kar et al. [59], Petiwala et al. [60], and Ossikbayeva et al. [61] showing a significant decrease in proliferation and survival of PC-3 and DU-145 prostate cancer cells with CA treatment.

Additionally, we examined the androgen sensitive, LNCaP prostate cancer cells and found significant inhibitory effects in agreement with findings by Petiwala et al. [60], while LNCaP cells lack the PTEN tumor suppressor, they are androgen responsive and therefore susceptible to cancer treatments such as androgen deprivation therapy [73]. Here we show that LNCaP cells are more susceptible to CA treatment with a lower IC_50_ compared to PC-3 cells. Petiwala et al. [60] found that treatment of LNCaP cells with CA resulted in significant androgen receptor (AR) degradation, and therefore this inhibitory effect on AR signaling may explain why LNCaP cells are more suspectable to CA treatment compared to the androgen-independent PC-3 cells. On the other hand, PNT1A cells, which have no known mutations and represent normal prostate epithelium were seen to have an IC_50_ 2-fold and 6-fold higher than PC-3 and LNCaP cells, respectively. These data indicate that CA concentrations that cause major inhibitory effects in prostate cancer cells have little toxicity to normal cells. Although, in vivo animal studies are required to examine the effects of CA in normal/healthy tissues, the present data suggest that CA may preferentially target cancer cells while sparing normal/healthy cells.

The PC-3 prostate cancer cells are androgen independent, lack the tumor suppressor, p53 [74], and the tumor suppressor (PTEN) leading to over activation of the PI3K-Akt cascade resulting in enhanced cell survival and proliferation [75,76,77]. Our studies show a decrease in phosphorylation/activation of Akt in PC-3 cells with CA treatment. Our findings are in agreement with the data by Kar et al. [59], who found inhibition of Akt phosphorylation, paired with increased phosphatase 2A (PP2A) activity in PC-3 cells treated with CA [59]. Whether the inhibition of Akt, seen in our study, is due to increased PP2A activity remains to be examined in future experiments. Similar to our data, carnosic acid has been shown to inhibit Akt phosphorylation/activation in hepatoma [78], lung cancer [79] and gastric cancer cells [80]. Studies have shown that Akt is elevated in approximately 70–100% of advanced cases of prostate cancer [81,82] and therefore chemicals such as CA that target Akt may hold a significant therapeutic potential.

Activated Akt leads to downstream phosphorylation and activation of mTOR and p p70 S6K resulting in increased protein synthesis and cell proliferation [83]. Our study is the first to show a decrease in phosphorylation/activation of mTOR and p70 S6K with CA treatment in PC-3 prostate cancer cells. While no other studies have examined the effects of CA treatment on prostate cancer cell mTOR and p70 S6K, CA inhibited mTOR in hepatoma [78], lung cancer [79] and gastric cancer cells [80]. In PC-3 prostate cancer cells mTOR and p70 S6K was inhibited with RE [58] and carnosol (COH) [84] treatment.

Furthermore, in the present study, we saw a robust phosphorylation of AMPK on Thr172, an established marker of its activation [38] with CA treatment. Although, we did not measure AMPK activity directly, the robust phosphorylation of acetyl-CoA carboxylase (ACC), a downstream target of AMPK, confirms AMPK activation. Phosphorylation of ACC by AMPK inhibits its activity resulting in inhibition of fatty acid synthesis and promotion of fatty acid oxidation [36,37]. Fatty acid synthesis is very important in sustaining cancer cell bioenergetic requirements and proliferation, and based on evidence of the key role of ACC in the regulation of fatty acid synthesis, and its overexpression in cancer cells, ACC has emerged as an attractive target for cancer treatment [85,86,87]. The inhibition of ACC, seen in our study, may contribute to inhibition of prostate cancer cell proliferation and survival.

As mentioned above, activation of Akt leads to activation of mTOR-p70 S6K signaling whereas inhibition of Akt leads to the inhibition of mTORC1-p70 S6K [23,24]. On the other hand, activation of AMPK phosphorylates/inhibits raptor, the regulatory protein associated with mTOR in the mTORC1 complex, resulting in mTOR inhibition [88]. In the present study, we found increased phosphorylation of raptor suggesting inhibition of the mTORC1 complex with CA treatment. Therefore, the inhibition of mTORC1-p70 S6K signaling, seen with CA treatment in the present study, could be due to both inhibition of Akt and activation of AMPK. These data are in agreement with previous studies, that found significant increase in AMPK phosphorylation/activation associated with significant phosphorylation of raptor and inhibition of p70 S6K signaling in H1299 lung cancer cells with RE treatment [55].

To elucidate how CA causes phosphorylation/activation of AMPK, we examined its upstream regulator, sestrin-2. Sestrin-2 acts as a scaffold to facilitate the interaction between AMPK and its upstream kinase and tumor suppressor LKB1 [89,90,91]. This is the first study to show that CA treatment increases the levels of sestrin-2 in PC-3 prostate cancer cells. However, these data are well-aligned with our previous findings in lung cancer cells showing increased expression of sestrin-2 with CA treatment [57]. One study has shown that overexpression of sestrin-2 in human prostate cancer PC-3 cells significantly reduced their proliferation and sensitized them to radiation treatment [92]. It should be noted that sestrin-2 expression varies among different types of cancer and both a tumor suppressor and a tumor promoter role has been indicated in different studies [93]. Sestrin-2 expression could be induced by DNA damage in a p53-dependent manner or by increased ROS levels in a p53-independent mechanism [44,94]. PC-3 cells lack the tumor suppressor p53 and have low basal levels of sestrin-2. It is possible that CA exhibits antitumor effects in PC-3 cells (lacking p53 expression) by increasing sestrin-2 levels independent of p53, similar to the effects seen in H1299 lung cancer cells [57] and to the effects seen with quercetin treatment of HCT116 and HT-29 colon cancer cells [95]. Treatment of HCT116 and HT-29 colon cancer cells with quercetin increased sestrin-2 expression and induced apoptosis by a mechanism that was p53-independent but involved increased ROS levels [95].

AMPK may be activated directly by sestrin-2 [71,72], or by sestrin-2-LKB1 dependent mechanism [89]. LKB1 levels are significantly lower in prostate cancer tissue compared to healthy prostate tissue, and PC-3 prostate cancer cells express low levels of LKB1 [96]. Although we did not examine the effects of CA on LKB1, it is possible that CA increases LKB1 levels and/or activity as seen in lung cancer cells [57]. AMPK activation is implicated in inhibition of protein synthesis [35], induction of autophagy [97] and may regulate apoptosis [98] overall leading to anticancer effects [30,31,32,33].

The use of the known AMPK activator, AICAR resulted in a significant decrease in PC-3 cell proliferation in agreement with studies by others showing inhibition of prostate cancer cell proliferation by AICAR [99,100]. Furthermore, the use of wortmannin and rapamycin resulted in similar inhibition of prostate cancer proliferation as CA. It has been shown that inhibitors of Akt and mTOR significantly reduce proliferation of prostate cancer cells [101,102,103]. Currently Akt (MK2206) and mTOR (Temsirolimus) inhibitors are being evaluated in clinical trials for patients with various forms of cancer including prostate cancer (NCT01480154, 00919035). Our findings show that CA is able to inhibit both Akt and mTOR and have effects similar to clinically evaluated inhibitors of these pathways, providing strong rationale for continuing the examination of the effects of CA in prostate cancer.

Overall, our data show a significant decrease in prostate cancer cell survival and proliferation in association with inhibition of Akt-mTOR-p70 S6K and activation of sestrin-2-AMPK signaling with CA treatment (Figure 8).

## 5. Conclusions

In the present study, prostate cancer cells treated with CA showed a significant decrease in survival and proliferation associated with decreased levels of phosphorylated/activated Akt and increased levels of phosphorylated/activated AMPK. Downstream of AMPK, CA inhibited mTORC1-p70 S6K and ACC signaling. Future studies should aim to examine the effects of CA in vivo utilizing animals xenografted with prostate cancer cells.

## Figures and Tables

**Figure 1 nutrients-16-01257-f001:**
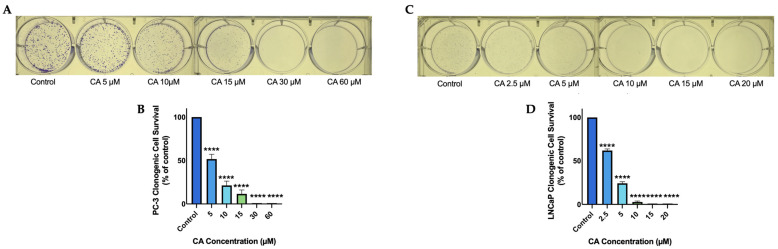
CA dose-dependently Inhibits prostate cancer cell survival. PC-3 (**A**,**B**) and LNCaP (**C**,**D**) prostate cancer cells were treated without (Control) or with the indicated concentrations of carnosic acid (CA) for 7 days followed by staining with methylene blue and colony counting Representative images were taken using a 12 MP camera (**A**,**C**). Data are the mean ± SEM of 3–4 independent experiments. **** *p* < 0.0001.

**Figure 2 nutrients-16-01257-f002:**
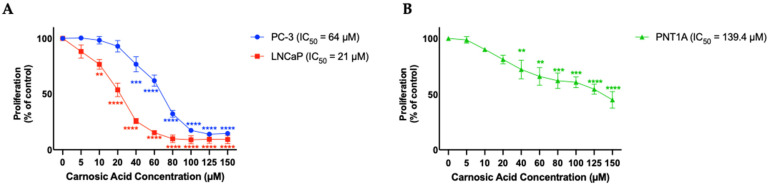
Effects of CA on Prostate Cancer cells (**A**) and PNT1A epithelial cells (**B**). PC-3 and LNCaP prostate cancer cells (**A**) and PNT1A prostate epithelial cells (**B**) were treated without (Control) or with the indicated concentrations of carnosic acid (CA) followed by fixing with 10% formalin and stained with 0.5% crystal violet dye. Crystal violet was solubilized, and absorbance was read at 570 nm. The data are the mean ± SEM of 4–6 independent experiments.** *p* < 0.01, *** *p* < 0.001, **** *p* < 0.0001.

**Figure 3 nutrients-16-01257-f003:**
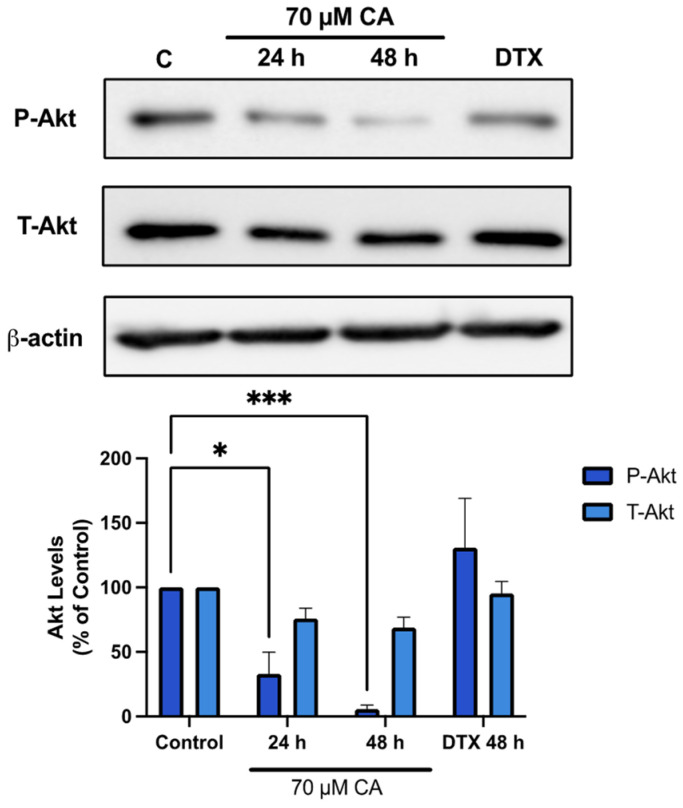
Carnosic acid inhibits Akt. PC-3 cells were treated without (Control) or with the indicated concentrations of CA or DTX for 24 or 48 h followed by whole cell lysate preparation and total protein yield assessment. Lysates (20 µg of protein) were resolved by SDS-PAGE and immunoblotted with specific antibodies against total or phosphorylated Akt (Ser473) or β-actin. Arbitrary units were used to express densitometry of bands using ImageJ software and the data are expressed as percent of control. The data are the mean ± SEM of 4–6 independent experiments. * *p* < 0.05, *** *p* < 0.001.

**Figure 4 nutrients-16-01257-f004:**
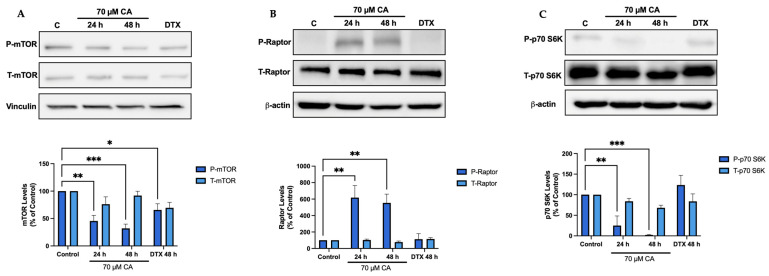
Carnosic acid inhibits mTOR-p70 S6K signaling. (**A**) mTOR phosphorylation. (**B**) raptor phosphorylation. (**C**) p70 S6K phosphorylation. PC-3 cells were treated without (Control) or with the indicated concentrations of CA or DTX for 24 or 48 h followed by whole cell lysate preparation and total protein yield assessment. Lysates (20 µg of protein) were resolved by SDS-PAGE and immunoblotted with specific antibodies against total or phosphorylated mTOR (Ser2448), raptor (Ser972), p70 S6K (Thr389), vinculin or β-actin. Arbitrary units were used to express densitometry of bands using ImageJ software and the data are expressed as percent of control. The data are the mean ± SEM of 2–6 independent experiments. * *p* < 0.05, ** *p* < 0.01, *** *p* < 0.001.

**Figure 5 nutrients-16-01257-f005:**
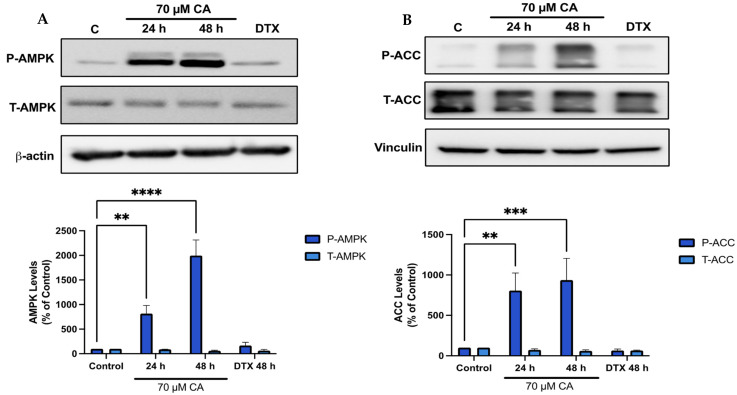
Carnosic acid activates AMPK. PC-3 cells were treated without (Control) or with the indicated concentrations of CA or DTX for 24 or 48 h followed by whole cell lysate preparation and total protein yield assessment. Lysates (20 µg of protein) were resolved by SDS-PAGE and immunoblotted with specific antibodies against total or phosphorylated AMPK (Thr172), ACC (Ser79), β-actin or vinculin. (**A**) AMPK phosphorylation/activation. (**B**) ACC phosphorylation/inhibition. Arbitrary units were used to express densitometry of bands using ImageJ software and the data are expressed as percent of control. The data are the mean ± SEM of 4–6 independent experiments. ** *p* < 0.01, *** *p* < 0.001, **** *p* < 0.0001.

**Figure 6 nutrients-16-01257-f006:**
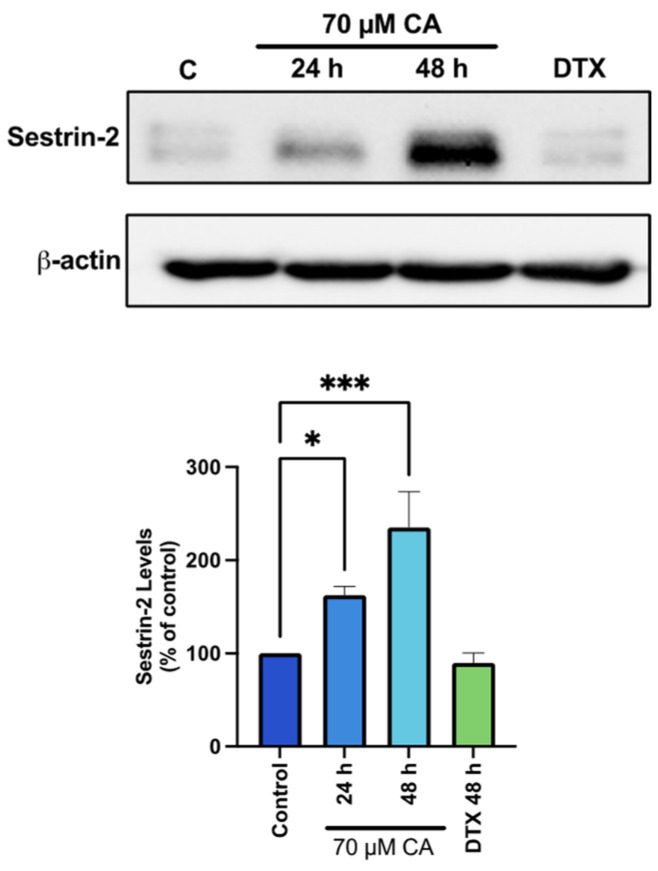
Activation of sestrin-2 by carnosic acid. PC-3 cells were treated without (Control) or with the indicated concentrations of CA or DTX for 24 or 48 h followed by whole cell lysate preparation and total protein yield assessment. Lysates (20 µg of protein) were resolved by SDS-PAGE and immunoblotted with specific antibody against total Sestrin-2 or β-actin. Arbitrary units were used to express densitometry of bands using ImageJ software and the data are expressed as percent of control. The data are the mean ± SEM of 3–5 independent experiments. * *p* < 0.05, *** *p* < 0.001.

**Figure 7 nutrients-16-01257-f007:**
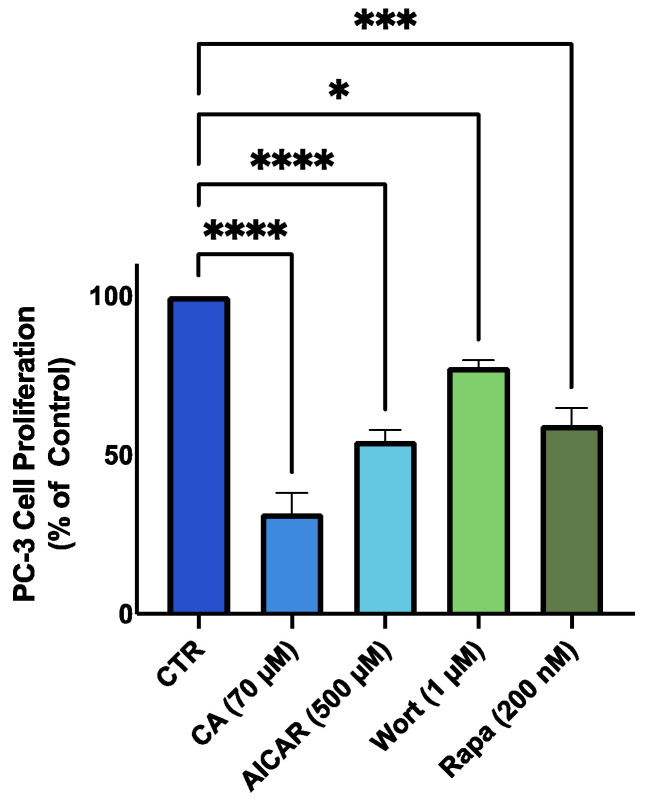
Effects of CA, AICAR, wortmannin and rapamycin on PC-3 prostate cancer cells. PC-3 cells were treated without (Control) or with 70 µM of carnosic acid (CA), 500 µM 5-Aminoimidazole-4-carboxamide ribonucleotide (AICAR) (AMPK activator), 1 µM wortmannin (Wort) (Akt inhibitor), or 200 nM rapamycin (Rapa) (mTOR inhibitor) for 48 h followed by fixing with 10% formalin and stained with 0.5% crystal violet dye. Crystal violet was solubilized, and absorbance was read at 570 nm. The data are the mean ± SEM of 2–3 independent experiments. * *p* < 0.05, *** *p* < 0.001, **** *p* < 0.0001.

**Figure 8 nutrients-16-01257-f008:**
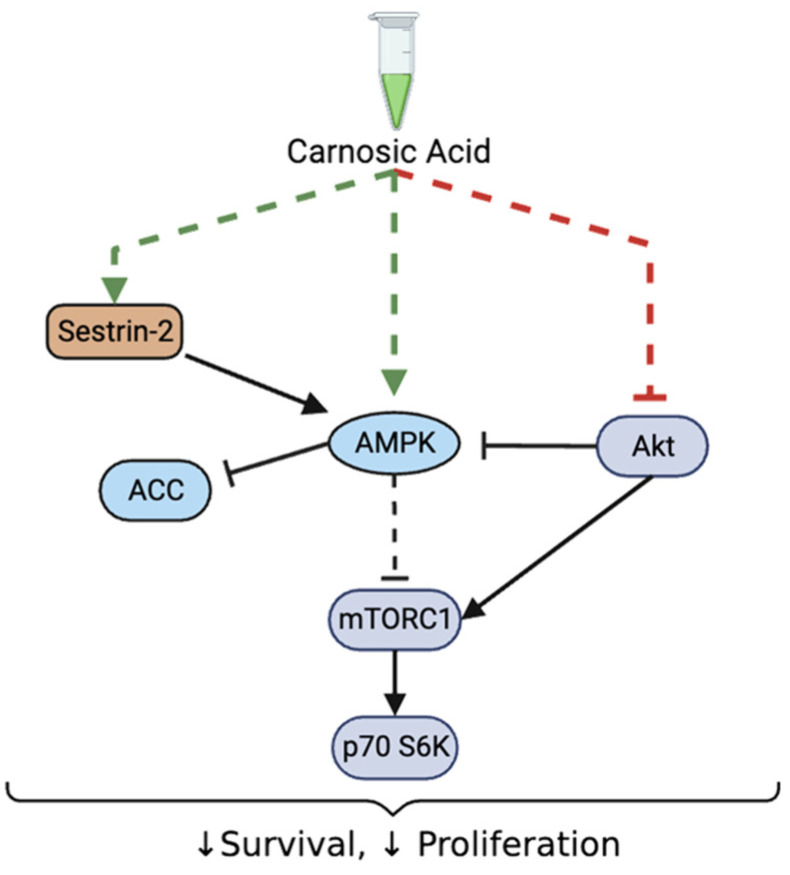
Summary of the effects of CA on prostate cancer cells. Carnosic acid decreased survival and proliferation of prostate cancer cells. These effects were associated with decreased Akt, mTOR and p70 S6K, increased total sestrin-2 levels and increased phosphorylation of raptor, AMPK, and ACC.

## Data Availability

Data are contained within the article.

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
