# Peer review of "Inhibition of Prostate Cancer Cell Survival and Proliferation by Carnosic Acid Is Associated with Inhibition of Akt and Activation of AMPK Signaling"

_nutrients, 2024, doi:10.3390/nu16091257_

Round 1
Reviewer 1 Report
Comments and Suggestions for Authors
The authors discussed the pressing need for more effective therapeutic agents targeting aggressive and hormone-resistant forms of PCa while sparing healthy cells. Then suggested carnosic acid (CA), a phenolic diterpene found in rosemary, as a potential anticancer agent. The manuscript is well written, the statistics are fine. However, I have the following comments:
- I suggest adjusting the p-value using the Benjamin Hochberg test (FDR).
- It would be great to have a KEGG pathway analysis of the findings.
-If possible, analyze the survival using one of the Kaplan-Meier or COX proportional analyses.
Author Response
Reviewer 1
The authors discussed the pressing need for more effective therapeutic agents targeting aggressive and hormone-resistant forms of PCa while sparing healthy cells. Then suggested carnosic acid (CA), a phenolic diterpene found in rosemary, as a potential anticancer agent. The manuscript is well written, the statistics are fine. However, I have the following comments:
- I suggest adjusting the p-value using the Benjamin Hochberg test (FDR).
- It would be great to have a KEGG pathway analysis of the findings.
-If possible, analyze the survival using one of the Kaplan-Meier or COX proportional analyses.
We thank this reviewer for the comments, and we are pleased that our manuscript was found to be well written.
The statistics used in the present manuscript is what is typically used to analyze data derived from in vitro cell culture studies. The stats that are suggested by this reviewer are used in the analysis of human clinical data and therefore cannot be applied here. For example, the Kaplan-Meier or COX proportional analyses are used for patient survival in clinical studies. Similarly, the Benjamini-Hochberg procedure, also known as the False Discovery Rate (FDR) procedure, is used in multiple hypothesis testing and does not apply here.
Reviewer 2 Report
Comments and Suggestions for Authors
The manuscript delivers an in-depth analysis of carnosic acid's (CA) anticancer activities against PC-3 prostate cancer cells, shedding light on its potential modes of action by affecting crucial signaling pathways. The research appears to be methodically conducted, offering pivotal insights into CA's therapeutic possibilities for treating prostate cancer. The detailed exploration of CA's impact on cell survival, proliferation, and significant signaling pathways including Akt, mTOR, p70 S6K, and AMPK highlights the meticulous efforts of the authors. Nonetheless, several aspects must be improved for a more comprehensive understanding and broader implications of the study:
1st, Rationale Behind Cell Line Choice and Broadening the Spectrum of Cell Lines: The selection of PC-3 cells, noted for their androgen independence and aggressive characteristics, is relevant for exploring treatments against more resilient prostate cancer forms. Yet, the manuscript could be enhanced by a deeper rationale for exclusively using this cell line. Elaborating on the genetic and morphological qualities that render PC-3 cells an apt choice for this study would solidify the basis for their use. Additionally, broadening the range of cell lines under investigation could augment the applicability of the findings. Incorporating a variety of cell lines, such as additional androgen-independent lines (DU-145/22RV1), androgen-sensitive prostate cancer cells (e.g., LNCaP/MycCaP), and non-cancerous prostate epithelial cells (BPH-1/RWPE-1), would provide invaluable insights into CA's specificity and potential adverse effects. Evaluating CA's influence across diverse prostate cancer models could yield a more rounded view of its therapeutic validity and safety.
2, Enhanced Comparative Analysis and Control Groups: While the study outlines CA's effects on PC-3 cells, juxtaposing these effects with those elicited by established Akt/mTOR inhibitors could further clarify CA's therapeutic efficacy and specificity. Moreover, investigating the role of the Sestrin-2-LKB1-AMPK axis in the context of CA-induced effects, particularly considering its crucial role in AMPK activation and diminished expression in prostate cancer, could deepen our understanding of CA's therapeutic promise.
3, Clarity on Experimental Reproducibility and Conditions: Although the manuscript comprehensively details experimental designs and outcomes, including CA's influence on cell proliferation/survival and modulation of signaling pathways, additional specifics could significantly aid in study replication and validation. Detailed descriptions concerning the CA concentration rationale, a solvent used, and illustrative images from assays such as colony formation would immensely contribute to the study's reproducibility and the integrity of its findings.
In conclusion, while the manuscript substantially contributes to our comprehension of carnosic acid's anticancer potential in prostate cancer, attending to the suggestions could markedly elevate the study's comprehensiveness and significance. Implementing these enhancements would not only fortify the current research but also set a more solid foundation for future inquiries, potentially accelerating the advancement of more efficacious cancer treatments.
Comments on the Quality of English Language
N/A
Round 2
Reviewer 2 Report
Comments and Suggestions for Authors
The reviewer acknowledges the extensive efforts the authors have made to address the questions and improve the manuscript. All the questions have been well addressed by providing additional data.
One suggestion to be considered in the following study: The author observed a stronger inhibitory effect in LNCaP over PC3. As the authors discussed, it would be worth looking into the mechanism of carnosic acid in androgen-responsive prostate cancer models, particularly determining the effect of carnosic acid on the androgen receptor signaling pathway.
According to the literature https://academic.oup.com/carcin/article/37/8/827/1744629, carnosic acid promotes the degradation of the androgen receptor. Please add more explanations and discussions on androgen-dependent prostate cancer for a broader reader's interest.
The manuscript is recommended to be accepted upon minor revision in the Discussion section.
Author Response
Thank You for this excellent suggestion. We have revised our manuscript accordingly.